# Secreted HSP90α-LRP1 Signaling Promotes Tumor Metastasis and Chemoresistance in Pancreatic Cancer

**DOI:** 10.3390/ijms23105532

**Published:** 2022-05-16

**Authors:** Nina Xue, Tingting Du, Fangfang Lai, Jing Jin, Ming Ji, Xiaoguang Chen

**Affiliations:** 1State Key Laboratory of Bioactive Substance and Function of Natural Medicines, Institute of Materia Medica, Chinese Academy of Medical Sciences and Peking Union Medical College, Beijing 100050, China; angelnina@imm.ac.cn (N.X.); ninadu@imm.ac.cn (T.D.); laifangfang@imm.ac.cn (F.L.); rebeccagold@imm.ac.cn (J.J.); 2Beijing Key Laboratory of Non-Clinical Drug Metabolism and PK/PD Study, Institute of Materia Medica, Chinese Academy of Medical Sciences and Peking Union Medical College, Beijing 100050, China

**Keywords:** secreted HSP90α, low-density lipoprotein receptor-related protein 1, epithelial-to-mesenchymal transition, metastasis, chemo-resistance, pancreatic cancer

## Abstract

The extracellular heat shock protein 90α (eHSP90α) has been reported to promote cancer cell motility. However, whether pancreatic cancer (PC) cells expressed membrane-bound or secreted HSP90α, as well as its underlying mechanism for PC progression, were still unclear. Our study demonstrated that the amounts of secreted HSP90α proteins were discrepant in multiple PC cells. In addition, highly invasive Capan-2 cells have a higher level of secreted HSP90α compared with those of less invasive PL45 cells. The conditioned medium of Capan-2 cells or recombinant HSP90α treatment stimulated the migration and invasion of PC cells, which could be prevented with a neutralizing anti-HSP90α antibody. Furthermore, secreted HSP90α promoted elements of epithelial–mesenchymal transition in PL45 cells, including increases in vimentin and Snail expressions, decreases in E-cadherin expression, and changes in cell shape towards a mesenchymal phenotype, but these phenomena were reversed by the anti-HSP90α antibody in Capan-2 cells. In addition, high levels of low-density lipoprotein receptor-related protein 1 (LRP1) were associated with worsened patient survival in pancreatic adenocarcinoma. We demonstrated LRP1 as a receptor of eHSP90α for its stimulatory role in metastasis, by activating the AKT pathway. In addition, silencing LRP1 enhanced the chemosensitivity to gemcitabine and doxorubicin in Capan-2 cells. Therefore, our study indicated that blocking secreted HSP90α underlies an aspect of metastasis and chemoresistance in PC.

## 1. Introduction

Pancreatic cancer (PC), one of the most devastating human malignancies, is reported to be the fourth leading cause of cancer death in the developed world. Moreover, PC patients show severe resistance to the current chemotherapeutic regimens and develop local recurrence and/or distant metastases following surgery. These result in the limitation of treatment efficacy. The currently reported overall five-year survival rate for PC is less than 11% [1]. The process of epithelial–mesenchymal transition (EMT) is suggested to be vital for tumor metastasis, drug resistance, and therapeutic perspectives. As described by Gaianigo N and Zhou PT et al., poor prognosis, owing to metastatic dissemination in an early event in PC, is mainly attributed to EMT [2,3]. The process of EMT is associated with the phenotypic conversion of epithelial cells into mesenchymal-like cells in cell culture conditions, which is characterized by a loss of cell–cell adhesion (with cobblestone-like appearance) and the acquisition of motile polarity (with scattered spindle-shaped morphology). This process is generally accompanied by changes in genome and protein levels, including epithelial markers (N-cadherin and ZO-1) and mesenchymal markers (N-cadherin, vimentin, Twist, and Snail, etc.) [4].

The heat shock protein 90α (HSP90α), an essential intracellular molecular chaperone, regulates protein folding, assembly, and maturation, to maintain cell survival and growth [5]. In addition to its intracellular function, HSP90α is also expressed on the cell surface and constitutively secreted by many types of cancer, including colorectal cancer, prostate cancer, breast cancer, and glioblastoma [6,7]. In these instances, it is known as extracellular HSP90α (eHSP90α), promoting cell motility and invasion, but eHSP90α does not appear in normal tissue in physiological conditions [8,9]. Thereby, selectively targeting eHSP90α would be more effective in tumor cells with less toxicity than the current inhibitors, which focus on intracellular HSP90α. Among all the studies on eHSP90 targets, low-density lipoprotein receptor-related protein (LRP1, also known as CD91), as the first reported surface receptor for eHSP90α, has been researched in relatively more detail [10]. Several studies have reported that, upon binding with LRP1, secreted HSP90α promotes the migration and invasion of various cancer cells through NF-κB, PI3K/AKT, or the MAPK-dependent pathway [11,12]. With the development of protein–protein interaction technology, several other proteins that interact with eHSP90α have been identified, such as the Toll-like receptor family (TLRs), the epidermal growth factor receptor family (EGFRs), and annexin Ⅱ [13,14]. For example, Sidera K. et al. reported that surface HSP90/HER-2 interactions lead to cytoskeletal rearrangement and cell motility in breast cancer. A study from Thuringer D. et al. showed that secreted HSP90α favors the cell migration of glioblastoma cells through TLR4-mediated EGFR activation. However, whether and how eHSP90α regulates the migration and invasion of PC have not been elucidated.

In this study, we demonstrated that secreted HSP90α enhanced the migration and invasion of PC cells. Upon binding to membrane receptor LRP1, secreted HPS90α’s elite AKT-dependent pathway regulates the mesenchymal-like morphology and marker gene expressions (Snail and vimentin), leading to PC cell metastasis. In addition, silencing LRP1 could increase drug sensitivity to gemcitabine (GEM) and doxorubicin (ADM) in Capan-2 cells. Our findings suggested that blocking secreted HSP90α presents an effective therapeutic agent for the treatment of metastasis and chemoresistance of PC.

## 2. Results

### 2.1. Secreted HSP90α Is Elevated in More Aggressive PC Cells 

Our studies were designed to investigate whether extracellular HSP90α (eHSP90α), including the membrane-bound and secreted HSP90α, were expressed on multiple PC cells. Membrane proteins were effectively isolated from six kinds of human cultured PC cells (Bxpc3, Mia-paca2, Capan2, SW1990, PL45, and PANC1), and the surface HSP90α levels were detected by Western blotting analysis. As shown in Figure 1A, Bxpc3 and PANC1 cells, by contrast, had lower HSP90α expression on the cell surface. The flow cytometry data indicated that the surface HSP90α expressions were low in these PC cells (Appendix A). However, the amounts of secreted HSP90α proteins were discrepant in the above-mentioned PC cells. Capan-2 cells secreted an amount of HSP90α protein, yet PL45 cells almost did not produce eHSP90α in the supernatant of tumor cells (Figure 1B). Additionally, Capan-2 cells were more invasive than PL45 cells (Appendix A). These results suggest that the amount of secreted HSP90α correlated more strongly with the PC cells’ invasion capacity than that of membrane-bound HSP90α. 

### 2.2. Secreted HSP90α Promoted PC Cell Migration and Invasion 

The stimulatory role of secreted HSP90α in PC cell migration was investigated using a wound-healing assay. We found that the conditional medium (CM) of Capan-2 cells or 10 μg/mL recombinant HSP90α (rHSP90α) protein treatment significantly induced the migration of PL45 and Capan-2 cells. Additionally, the CM-induced cell migration activity was inhibited by the anti-HSP90α antibody (Figure 2A,B). Next, the invasive ability of Capan-2 and PL45 cells was examined using a Transwell assay. As shown in Figure 2C,D, Capan-2 cells were more invasive than PL45 cells. Treatment with the rHSP90α protein could promote the invasion of Capan-2 and PL45 cells, while HSP90α neutralizing antibody treatment significantly inhibited the invasion activities of Capan-2 cells. It is important to note that the effects of the rHSP90α and the anti-HSP90α antibody used in these assays are directed toward cell migration and invasion, not toward cell proliferation, since the absorbances_λ570_ showed almost no apparent differences in the Capan-2 or PL45 cultures treated as described above (Appendix A). These results suggest that secreted HSP90α is responsible for the induction of PC cell migration and invasion.

### 2.3. Secreted HSP90α Promoted PC Cell EMT

To examine whether secreted HSP90α functionally regulated PC cells’ motility, consistent with the activation of an EMT, we firstly detected the EMT hallmark molecules, such as E-cadherin and vimentin in the above-mentioned six types of PC cells. Interestingly, these E-cadherin over-expressing cells, including SW1990, Bxpc3, and PL45, hardly expressed vimentin, while Capan-2, Mia-paca2 and PANC1 cells with high levels of vimentin barely expressed E-cadherin, indicating that E-cadherin and vimentin molecules could represent counteractors of EMT markers (Figure 3A). In light of this, we chose Capan-2 and PL45 as the cell pair for monitoring EMT events. In our experiments, we found that CM or rHSP90α treatment could induce vimentin expression and down-regulate E-cadherin levels in PL45 cells. Moreover, treatment with the anti-HSP90α antibody could reverse the up-regulated vimentin expression induced by CM (Figure 3B–D). Additionally, epithelial-like PL45 or SW1990 cells exhibited mesenchymal phenotypes after treatment with CM of Capan-2 or rHSP90α, such as the tightly packed epithelial cobblestone pattern conversion to a scattered spindle-shaped phenotype (Figure 3E; Appendix A). In the presence of the anti-HSP90α antibody, the mesenchymal phenotype induced by CM had the tendency to change to a contractile phenotype (Figure 3E). In addition, we also observed that the mRNA levels of mesenchymal molecules (such as N-cadherin, vimentin or Snail) were significantly increased in CM or rHSP90α-treated PL45 cells, whereas the mRNA level of the epithelial marker E-cadherin notably decreased after CM treatment (Figure 3F,G). Furthermore, the latter event was reversed in combination with the anti-HSP90α antibody (Figure 3F).

### 2.4. Neutralization of Secreted HSP90α Induced PC Cell MET

Targeting secreted HSP90α in mesenchymal-like Capan-2 cells by the anti-HSP90α neutralizing antibody resulted in a tight cellular cluster resembling the epithelial phenotype (Figure 4A). Furthermore, the high levels of vimentin or Snail were repressed by the anti-HSP90α antibody and DMAG-N-oxide (NPGA) treatment, especially by the HSP90α neutralizing antibody (Figure 4B,C). The Snail and vimentin mRNA levels were significantly reduced after exposure to the NPGA or anti-HSP90α antibody (Figure 4D). These data indicate that targeting secreted HSP90α in the mesenchymal phenotype could reverse EMT events, resulting in MET conversion.

### 2.5. Secreted HSP90α Acts through LRP1 Signaling to Regulate PC Cell EMT

LRP1 is the first reported cell membrane receptor for extracellular HSP90α. Previous studies reported that high LRP1 expression was related to tumor cell migration, invasion, or poor prognosis, such as in breast cancer, glioblastoma, and pancreatic ductal adenocarcinoma [15,16,17]. Using TCGA datasets, we found that increased LRP1 expression significantly correlated with decreased patient survival in pancreatic cancer (*p* = 0.018, n = 176) (Figure 5A). The correlations of LRP1 mRNA expressions and pro-EMT transcription factors were analyzed in data extracted from the TCGA. In this cohort, the correlation analysis revealed that LRP1 mRNA expression was moderately to strongly correlated with the Snail mRNA level (Pearson *r* =0.63, *p* < 0.01) (Figure 5B). In our experiments, we used targeted shRNA to effectively knockdown the mRNA and protein levels of LRP1 in Capan-2 cells (Figure 5C). Silencing LRP1 significantly decreased the Snail and vimentin protein expressions, with a slight increase in E-cadherin in Capan-2 cells (Figure 5D). Furthermore, the mRNA levels of mesenchymal markers (Snail and vimentin) and the invasion ability of Capan-2 cells were remarkably reduced after the down-regulation of LRP1 (Figure 5E,F). rHSP90α treatment significantly decreased the E-cadherin mRNA levels in Capan-2 cells, but increased the N-cadherin and Snail mRNA levels, whereas these effects of rHSP90α on the above-mentioned genes were diminished after silencing LRP1 in Capan-2 cells (Figure 5E). ERK and AKT, known as downstream signaling molecules of extracellular HSP90α, were activated after treatment with the rHSP90α protein in Capan-2 cells (Figure 5G,I). Silencing LRP1 could strikingly decrease Akt phosphorylation, without obvious effects on phosphorylated ERK, suggesting that eHSP90α mainly acts through the LRP1-mediated AKT signaling pathway (Figure 5H,J).

### 2.6. Silencing LRP1 Could Overcome Chemo-Resistance in Metastatic PC Cells

We investigated the sensitivity of Capan-2 cells transfected with shLRP1 to the following chemotherapeutic agents: gemcitabine (GEM), doxorubicin (ADM), topotecan (TPT), and paclitaxel (PTX). The cell viability after treatment with these drugs for 72 h was determined by MTT assay. As shown in Figure 6A, GEM, ADM, and TPT could dose-dependently decrease the viability of Capan-2 cells, with the IC_50_ value ranging from about 10^−7^ to 10^−6^ M. Moreover, the silencing of LRP1 was observed to promote the chemosensitivity of Capan-2 cells to GEM and ADM, but not to TPT and PTX (Figure 6B–E). Down-regulated LRP1 of Capan-2 cells show an IC_50_ value of 0.127 μM and 0.777 μM, treated by ADM or GEM, significantly different to the 0.721 μM and 1.494 μM value for the Capan-2 control cells.

## 3. Discussion

Previous studies have reported that the blockade of extracellular HSP90α (eHSP90α) inhibited cell motility in several types of cancers [6,8,18]. However, a unifying mechanistic basis for the metastatic function of eHSP90α in PC has not yet been defined. Our study showed that the amount of secreted HSP90α correlated more strongly with the PC cells’ invasion capacity than that of membrane-bound HSP90α. Secreted HSP90α proteins were increased in metastatic Capan-2 cells, comparable to the levels observed in less invasive PL45 cells. Additionally, rHSP90α protein treatment enhanced the migration and invasion of PC cells, further underscoring that secreted HSP90α is a potent driver of metastasis in PC. Activation of the EMT program was considered a major driver of tumor progression from initiation to metastasis [19,20]. We investigated the EMT initiating activity, including several critical molecules and cell morphology consistent with this program to support the regulatory ability of secreted HSP90α in PC cells. Generally, epithelial markers include E-cadherin, γ-catenin, and zonula occludens-1 (ZO-1), whereas mesenchymal markers include the following: fibronectin, N-cadherin, β-catenin, vimentin, Snail, Twist, Zeb, etc. [4,6]. According to the level of EMT markers in several PC cell lines in our experiments, highly epithelial cell lines, as reported in [21,22], such as Bxpc3, have high levels of E-cadherin, while another group of PC cells (Mia-paca2 and PANC-1) exhibited strong expression of vimentin, indicating that E-cadherin and vimentin could be used as a pair of counteracting markers for distinguishing the epithelial–mesenchymal phenotype. We found that secreted or ectopic HSP90α could facilitate EMT in PL45 cells, whereas HSP90α neutralizing antibody treatment could induce MET in Capan-2 cells, by regulating cellular E-cadherin and vimentin protein expression. Taken together, we suggest that secreted HSP90α-induced EMT contributes to the high propensity of metastasis in PC.

So far, several other proteins (TLRs and EGFRs) binding to eHSP90α have been reported, but LRP1, as the receptor of eHSP90α, is still mainly studied in cancer metastasis [9]. LRP1 is a ubiquitously expressed surface receptor with numerous ligands, used to regulate a wide range of biological functions [23]. LRP1 is highly expressed in multiple types of cancers, and its mRNA level was associated with patient survival in bladder urothelial carcinoma [24]. Our studies supported the idea that a high LRP1 mRNA level was associated with worsened patient survival, and LRP1 expression is significantly more highly correlated with the level of Snail in the PC cohort from the TCGA database. Snail is a zinc transcription factor that mediates EMT in tumor cells. As a repressive transporter of the E-cadherin promoter, Snail inhibition is likely to up-regulate E-cadherin expression, resulting in reversal of the EMT phenotype, a reduction in tumor cell migration and proliferation, and enhancement of drug sensitivity in PC [25,26,27]. Our data demonstrated that silencing LRP1 in PC cells not only suppresses metastasis, but also enhances chemo-sensitization to GEM and ADM, and these results may be related to the inhibition of the Snail-induced EMT program.

As reported, through the cell membrane receptor LRP1/CD91, eHSP90α activated PI3K/AKT, ERK or NF-κB signaling to promote cell motility [12,28]. Consistently, we found that secreted HSP90α stimulated AKT and ERK activation in PC cells. However, genetic silencing of LRP1 to attenuate eHSP90α signaling decreased the levels of phosphorylated AKT, without an effect on the phosphorylated ERK level. Thus, we suggested that eHSP90α-induced metastasis, through LRP1, is dependent on the AKT pathway in PC cells. However, the administration of rHSP90α protein in LRP1 knockdown Capan-2 cells still reduced the epithelial marker (E-cadherin) gene expression and increased the mesenchymal marker (Snail and vimentin) gene expressions, indicating that, except for the LRP1-AKT-dependent pathway, other interacting protein-mediated signaling would participate in the secreted HSP90α-induced EMT. The detailed mechanism still needs to be further explored.

Overall, our study highlights the important role of secreted HSP90α in the metastasis of PC. Upon binding to LRP1, secreted HSP90α promoted the migration and invasion of PC cells through the regulation of EMT marker proteins (vimentin and Snail) and cell morphology. Mechanically, the secreted HSP90α-induced EMT process was partially in a LRP1-AKT-dependent manner. Furthermore, down-regulation of LRP1 could enhance the chemosensitivity in Capan-2 cells. Therefore, our study reveals that intervention into the secreted HSP90α signaling could be a new therapeutic strategy for metastasis and chemoresistance in PC.

## 4. Materials and Methods

### 4.1. Cell Culture and Chemical Reagents

The human PC cell lines (Capan2, Mia-paca2, SW1990, Bxpc3, PANC1 and PL45) were purchased from the American Type Culture Collection (ATCC, Manassas, VA, USA) or Cell Culture Center at the Institute of Basic Medical Sciences, Chinese Academy of Medical Sciences. These cell lines were maintained in Dulbecco’s modified Eagle’s medium (DMEM, Gibco^®^, Life Technologies TM, Carlsbad, CA, USA) supplemented with 10% fetal bovine serum (FBS, Gibco), streptomycin (100 μg/mL) and penicillin (100 U/mL) at 37 °C in a humidified atmosphere containing 5% CO_2_. Topotecan (TPT), doxorubicin (ADM), gemcitabine (GEM) and paclitaxel (PTX) were obtained from Selleck (Selleck Chemicals, Shanghai, China). The anti-HSP90α neutralizing antibody was obtained from Enzo Life Sciences (Plymouth Meeting, PA, USA). E-cadherin, vimentin, Snail, LRP1, phospho-AKT_ser473_, AKT, phospho-ERK1/2_Thr202/Tyr204_, ERK and β-actin were purchased from Cell Signaling Technologies (Danvers, MA, USA). The antibody to HSP90α was obtained from Abcam (Cambridge, MA, USA).

### 4.2. Wound-Healing Assay

Capan-2 and PL45 cells were seeded in 6-well plates and cultured at 37 °C. After being left overnight, a wound was generated using a pipette tip to make a straight scratch. The cells were treated with or without 5 μg/mL of the anti-HSP90α antibody or human recombinant HSP90α (rHSP90α, 10 μg/mL) protein for 16 h. The cells were imaged in 4 random fields per well under a microscope. The width of the wound was analyzed using Image J software (National Institutes of Health, Bethesda, MD, USA). 

### 4.3. Transwell Assays

The invasive assay was performed as described, using Transwell cell culture chambers (8 μm pore size polycarbonate membrane; Costar, Cambridge, MA, USA). The membrane was coated with Matrigel (BD Biosciences, Bedford, MA, USA). Capan-2 and PL45 cell suspension without FBS (200 μL, 1 × 10^6^/mL cells) was placed in the upper chamber, while the bottom chamber was filled with 600 μL of culture medium containing 10% FBS. The anti-HSP90α antibody or rHSP90α protein (10 μg/mL) was added to the FBS-free medium in the upper chamber. Cells that migrated to the lower chamber were fixed with 4% paraformaldehyde for 20 min, followed by staining with 0.1% crystal violet (Solarbio, Beijing, China) for 10 min. The cells were randomly photographed under a light microscope. The migrated cell counts were calculated by Image J software.

### 4.4. Cell Survival Assays

MTT (3-(4,5-dimethylthiazol-2-yl)-2,5-diphenyl tetrazolium bromide) assay was adapted to measure cell viability using Capan-2 and PL45 cells. Cells were treated with different concentrations of GEM (0.03~20 μM), TPT (0.03~20 μM), PTX (0.03~20 μM) or ADM (0.003~3 μM) for 72 h. Then, MTT (0.5 mg/mL) solution was incubated for 4 h. The formazan crystals were solubilized in DMSO and measured at 570 nm using a microplate reader (Biotek Instruments, Inc., Winooski, VT, USA). The IC_50_ values were calculated using the GraphPad Prism 8 software (San Diego, CA, USA).

### 4.5. Quantitative RT-PCR (qRT-PCR)

Total RNA was extracted by the Easypure RNA kit (Tansgen Biotech, Beijing China). cDNA was obtained using the TransScript One-Step gDNA Remove and cDNA Synthesis SuperMix kit (Tansgen, Beijing China). PCR amplifications were performed using a SYBR Green PCR Master Mix kit (Cat. QPK-201, Toyobo, Japan) and an ABI PRISM 7900 Sequence Detection system (Applied Biosystems, Foster City, CA, USA). The primers used for qPCR were as follows: LRP1: F, 5′-GATGAGACACACGCCAACTG-3′; R, 5′-CGGCACTGGAACTCATCA-3′. E-cadherin: F, 5′-CAATGCCGCCATCGCTTAC-3′; R, 5′-ATGACTCCTGTGTTCCTGTTAATG-3′. N-cadherin: F, 5′-GACAATGCCCCTCAAGTGTT-3′; R, 5′-CCATTAAGCCGAGTGATGGT-3′. Vimentin: F, 5′-TCCGCACATTCGAGCAAAGA-3′; R, 5′-ATTCAAGTCTCAGCGGGCTC-3′. Snail: F, 5′-GCTCCACAAGCACCAAGAGT-3′; R, 5′-ATTCCATGGCAGTGAGAAGG-3′. GAPDH: F, 5′-GAGTCAACGGATTTGGTCGT-3′; R, 5′-TTGATTTTGGAGGGATCTCG-3′. GAPDH was used as an internal control. The indicated gene expression was calculated according to the 2^−ΔΔCT^ method.

### 4.6. Western Blot Assays

Treated and untreated PC cell lysates (30 μg) were subjected to SDS-PAGE and transferred to PVDF membranes (Millipore, Bedford, MA, USA). The membranes were blocked and incubated with the specific primary antibody (1:1000), followed by applying the corresponding HRP-conjugated secondary antibodies. The bands were visualized using ECL detection reagent, using Image Quant LAS 4000 (GE Healthcare, Piscataway, NJ, USA). The relative protein levels were calculated based on β-actin as the loading control and were densitometrically analyzed using Image J software. 

### 4.7. RNA Interference Assay

Short hairpin RNAs for the LRP1 gene were obtained from GenePharma Co., Ltd. (Shanghai, China). shLRP1 plasmid and control shRNA were transfected using Lipofectamine 3000 reagent (Thermo Fisher Scientific, Waltham, MA, USA), according to the manufacturer’s protocol. Stably transfected cells were selected with 1.5 μg/mL of puromycin. The LRP1 expression was confirmed by qPCR or Western blotting with a specific antibody.

### 4.8. Data Mining and Bioinformatic Analyses

LRP1 mRNA expression data from the PC dataset of The Cancer Genome Atlas (TCGA; https://tcga-data.nci.nih.gov, accessed on 12 November 2019) were analyzed using cBioportal for Cancer Genomics (http://cbioportal.org, accessed on 12 November 2019) web resources. RNA sequencing data (FPKM values) for gene expression were downloaded from the Genomic Data Commons (GDC, https://portal.gdc.cancer.gov/, accessed on 12 November 2019) using the R package TCGAbiolinks, and were transformed into transcripts per kilobase million (TPM) values, which are more similar to those resulting from microarrays and more comparable between samples [29]. We set the high and low gene expression level groups by the median value. Overall survival and Pearson’s correlation analyses for LRP1 and Snail were conducted to obtain gene expression level score.

### 4.9. Statistical Analysis

The results are expressed as the mean values ± SD. The statistical significance of the obtained data was calculated using Student’s *t* test. *p* < 0.05 was considered statistically significant. We used Kaplan–Meier estimation to analyze survival rates and the log-rank test to determine the significance of the differences between two survival curves. A Cox proportional hazards model was used for the univariate and multivariate analyses, and the hazard ratio was calculated using a 95% confidence interval (CI). Correlation coefficients between LRP1 and Snail mRNA expression were computed by Spearman and distance correlation analyses.

## Figures and Tables

**Figure 1 ijms-23-05532-f001:**
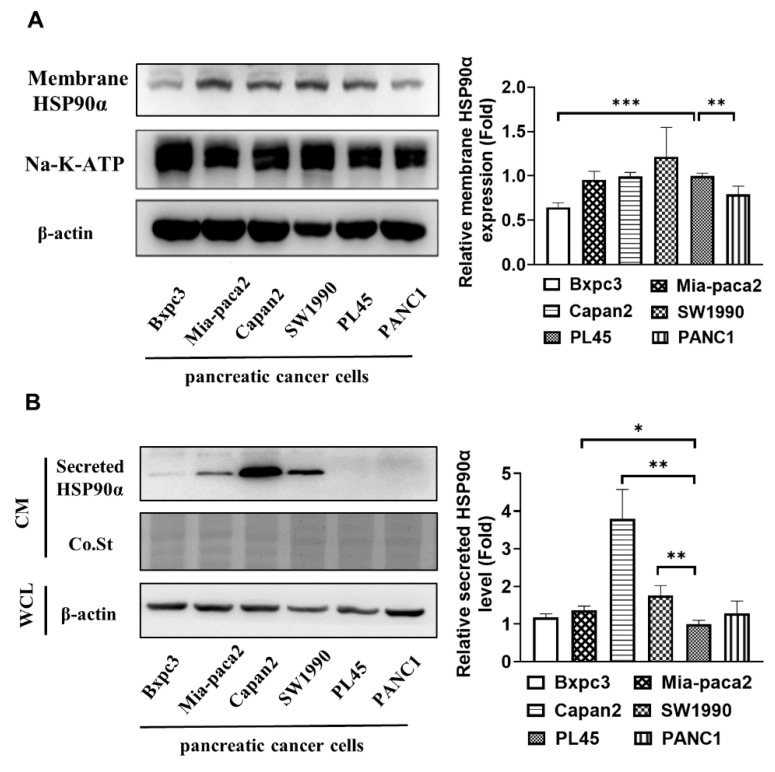
The levels of secreted HSP90α elevate in invasive PC cells. (**A**) A total of 5 × 10^6^ PC cells were plated overnight, then membrane proteins were isolated using a Thermo Scientific Mem-PER Plus membrane protein extraction kit, and the surface HSP90α level was detected by Western blot analysis. (**B**) A total of 5 × 10^6^ PC cells were seeded in a 10 cm dish with serum starvation for 24 h, and cell supernatants were filtered through 0.45 μM filters and concentrated by an Amicon Ultracel-30k centrifugal filter (Millipore) for equal volume. The secreted HSP90α levels were detected by Western blot analysis. The relative HSP90α protein levels in the cell surface or supernatant are shown in the histogram. CM: conditional medium, Co.St: Coomassie stain, and WCL: whole cell lysates. Data are expressed as the mean ± SD from three experiments. * *p* ≤ 0.05, ** *p* ≤ 0.01, and *** *p* ≤ 0.001.

**Figure 2 ijms-23-05532-f002:**
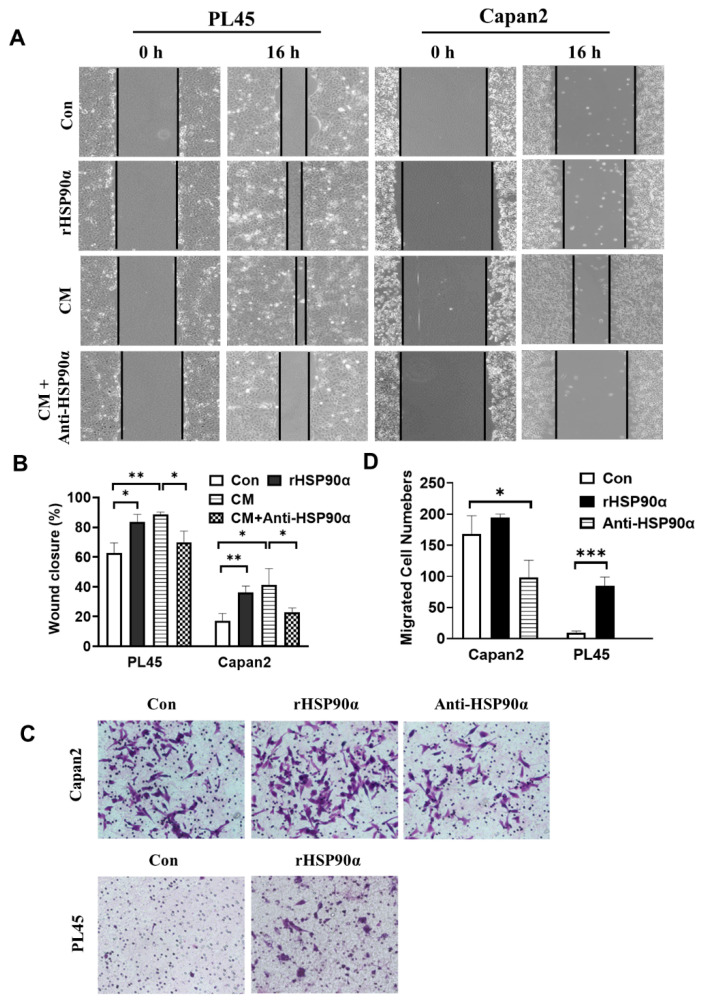
Secreted HSP90α induces the migration and invasion of PC cells. (**A**) Exogenous HSP90α increases the migration activities of PC cells. PL45 and Capan-2 cells were treated with 10 μg/mL of rHSP90α and CM with or without the anti-HSP90α antibody (5 μg/mL) for 16 h, and the migratory ability of cells was detected by wound-healing assays. (**B**) The wound closure rates at the indicated time points were shown as mean ± SD of three independent measurements. * *p* ≤ 0.05 and ** *p* ≤ 0.01. (**C**) rHSP90α increases cell invasiveness in PC cells. PL45 and Capan-2 cells were treated with PBS, rHSP90α (10 μg/mL) or the anti-HSP90α antibody (5 μg/mL) for 24 h and allowed to invade through Matrigel for 16 h. Invasive cells on the filters of the Transwell inserts were measured under a microscope. (**D**) The graphs show the quantitative evaluation of the migrated cell counts. Data are represented as the mean ± SD of three independent measurements. ** *p* ≤ 0.01 and *** *p* ≤ 0.001.

**Figure 3 ijms-23-05532-f003:**
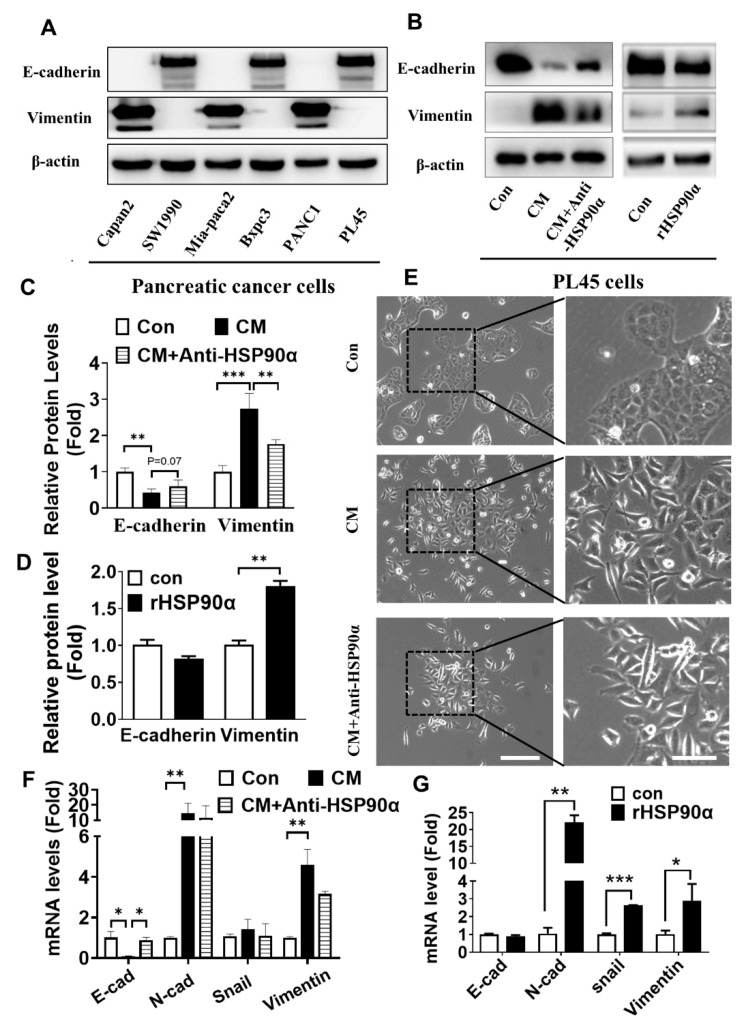
Secreted HSP90α induces epithelial–mesenchymal transition (EMT) in PC cells. (**A**) The protein expressions of EMT marker genes (E-cadherin and vimentin) were examined from six human PC cell lines. (**B**) The expressions of E-cadherin and vimentin in CM-treated or rHSP90α-treated PL45 cells, with or without the anti-HSP90α antibody (5 μg/mL), were detected by Western blot analysis. (**C**,**D**) The graphs show the quantitative evaluation of the relative protein levels. The results represent the averages of three independent experiments. ** *p* ≤ 0.01 and *** *p* ≤ 0.001. (**E**) The images of PL45 cells’ morphology after being stimulated by CM of Capan-2 or in the presence of the anti-HSP90α antibody for 72 h. Scale bar, 5 μm. (**F**,**G**) The relative mRNA levels of genes encoding E-cadherin, N-cadherin, vimentin, and Snail in CM, CM plus the anti-HSP90α antibody, or rHSP90α-treated PL45 cells were detected by qRT-PCR. The data represent the mean ± SD of three independent measurements. * *p* ≤ 0.05, ** *p* ≤ 0.01 and *** *p* ≤ 0.001.

**Figure 4 ijms-23-05532-f004:**
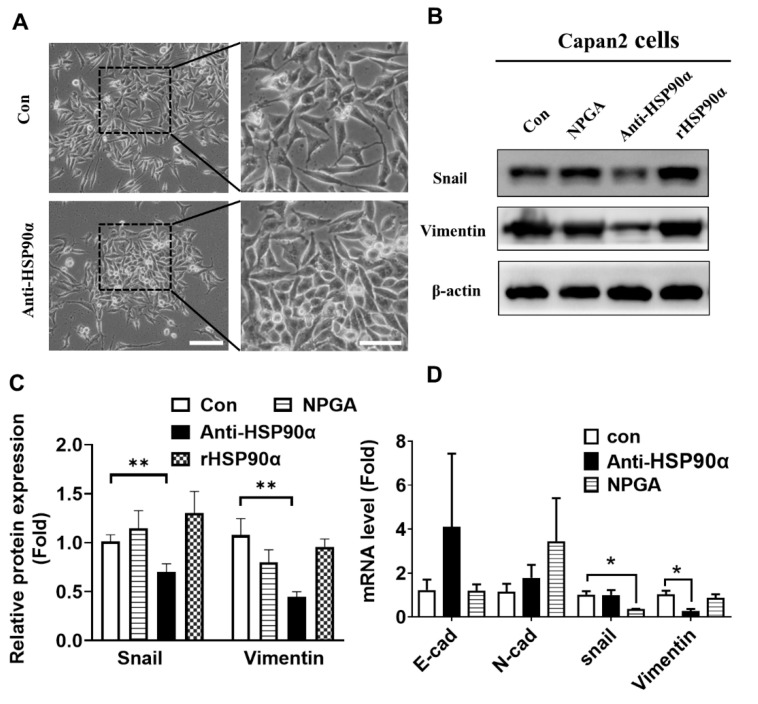
The anti-HSP90α neutralizing antibody reverses epithelial–mesenchymal transition (EMT) in PC cells. (**A**) The cell morphology after administration of the anti-HSP90α antibody (5 μg/mL) for 72 h in Capan-2 cells. Scale bar, 5 μm. (**B**) The immunoblot analysis of mesenchymal proteins, including vimentin and Snail, after treatment with the anti-HSP90α antibody or NPGA in Capan-2 cells. (**C**) The relative protein levels are shown in the histogram. Data are expressed as the mean from three experiments. ** *p* ≤ 0.01. (**D**) The E-cadherin, N-cadherin, vimentin, and Snail mRNA expressions in anti-HSP90α antibody-treated or NPGA-treated Capan-2 cells were validated by qRT-PCR. Data are expressed as the mean from three experiments. * *p* ≤ 0.05.

**Figure 5 ijms-23-05532-f005:**
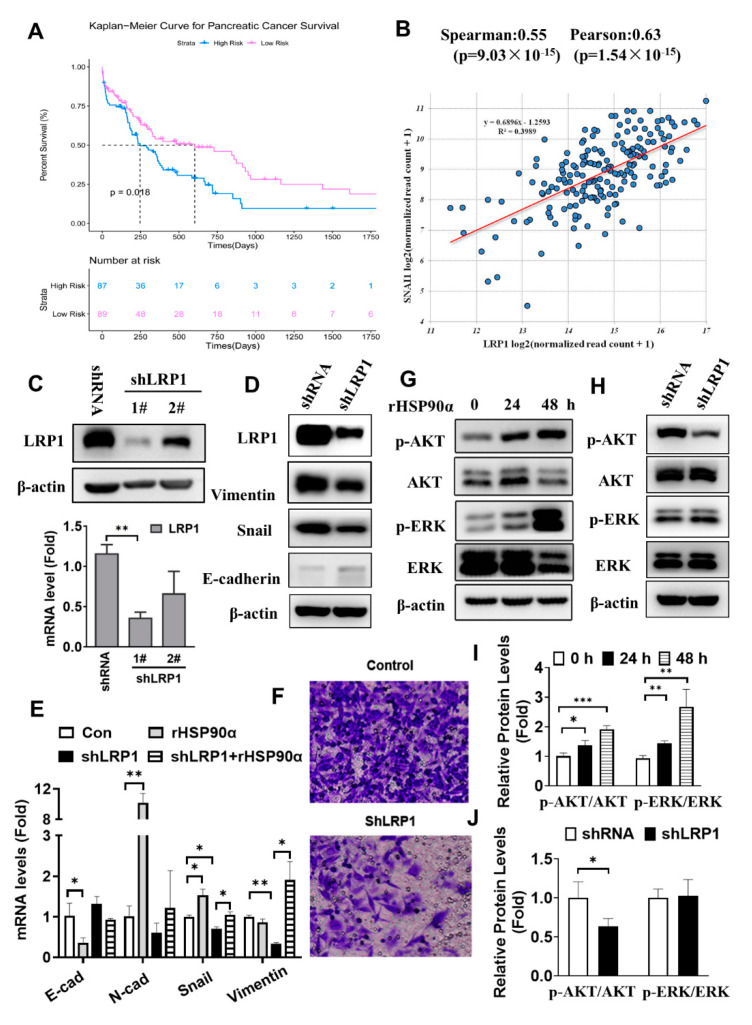
LRP1-medited AKT signaling is involved in secreted HSP90α-induced metastasis and EMT events in PC cells. (**A**) Kaplan–Meier curves of overall survival probability for low (red line) and high (blue line) LRP1 expression in 176 PC patients, using TCGA databases (log-rank test, *p* < 0.05). (**B**) Correlation analysis between the LRP1 mRNA expression and the Snail mRNA level in data extracted from the TCGA databases. (**C**) The LRP1 mRNA and protein level after LRP1 knockdown by vector-mediated RNAi. (**D**) The LRP1, vimentin, Snail, and E-cadherin protein expressions in stably transfected shLRP1 Capan-2 cells. (**E**) The mRNA levels of E-cadherin, N-cadherin, Snail and vimentin after stably transfecting shLRP1 in Capan-2 cells or in the presence of rHSP90α were detected by qRT-PCR. The data are expressed as the mean ± SD of the results from three separate experiments (* *p* < 0.05 and ** *p* < 0.01). (**F**) The invasion ability of Capan-2 cells after silencing LRP1 by Transwell assay. (**G**,**H**) The AKT and ERK signaling pathway activation in Capan-2 cells after rHSP90α treatment or LRP1 knockdown were detected by Western blot analysis. (**I**,**J**) The relative protein levels are shown in the histogram. The data are expressed as the mean from three experiments. * *p* ≤ 0.05, ** *p* ≤ 0.01 and *** *p* ≤ 0.001.

**Figure 6 ijms-23-05532-f006:**
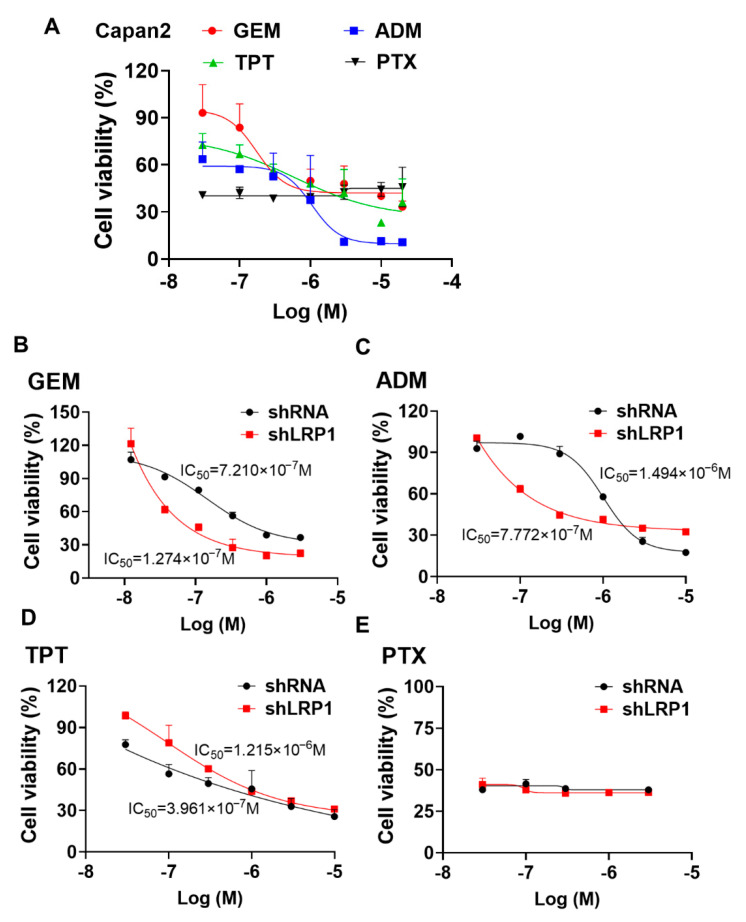
Silencing LRP1 could enhance chemo-sensitivity in PC cells. (**A**) The cell viability of Capan-2 was determined by MTT assay after exposure to different chemotherapeutic drugs for 72 h. The data are expressed as the mean ± SD of the results from three separate experiments. (**B**–**E**) The Capan-2 cells stably transfected control shRNA and targeted shLRP1 were administrated different concentrations of gemcitabine (GEM), doxorubicin (ADM), topotecan (TPT) or paclitaxel (PTX) for 72 h; the cell viability was detected by MTT assay.

## Data Availability

All data generated or analyzed during this study are included in this published article and its Appendix A.

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
