# Peer review of "Secreted HSP90α-LRP1 Signaling Promotes Tumor Metastasis and Chemoresistance in Pancreatic Cancer"

_ijms, 2022, doi:10.3390/ijms23105532_

Round 1

Reviewer 1 Report

All comments addressed. English proofreading still required.

Author Response

Thank you very much for these precious comments concerning my manuscript entitled “Secreted HSP90α-LRP1 signaling promotes tumor metastasis and chemoresistance in pancreatic cancer” (ijms-1701289).

These comments are all valuable and very helpful for revising and improving my paper.

We have carefully checked our manuscript and made corrections in revised tracked manuscript.

We hope meet with approval!

Best regards!

Reviewer 2 Report

Reviewer comments:

Comments to the Author

This manuscript describes the extracellular heat shock protein 90α (eHSP90α) promotes cancer cell motility. Authors explained that its expression is highly variable in different kinds of cancerous cell depending upon the aggressiveness and invasiveness. This study demonstrated that the amounts of secreted HSP90α proteins were discrepant in multiple PC cells as shown by highly invasive Capan2 cells have a higher level of secreted HSP90α, compared with those of less invasive PL45 cells.

The manuscript is impressive and for the most part well written with proper experimental design. Their discussion goes with the content provided according to the findings. The references are appropriate and timely.

Minor comments:

  • Whether the PL45 cell were checked before the Matrigel experiment because the cells don’t look healthy in the “Con” group.
  • Please provide better images for Figure 2C for PL45 “Con” group and include images for anti-HSP90α group.

Author Response

Thank you very much for these precious comments concerning my manuscript entitled “Secreted HSP90α-LRP1 signaling promotes tumor metastasis and chemoresistance in pancreatic cancer” (ijms-1701289).

These comments are all valuable and very helpful for revising and improving my paper.

Response: 

We make sure that the morphology of PL45 cells were right in our experiments. With a typical epithelial phenotype as showed in Figure 3E, the PL45 pancreatic cancer cell showed a weakly invasive ability in our Figure 2C and Figure 1SB. Figure 2C represents the invasive ability of PL45 cell. The migrated cell in “Con” group in PL45 cells is very few, the dyed round holes in the Figure 2C is the pores of chamber. In statistical analysis for migrated cell numbers, we set the size of particles to exclude the stained holes, and calculated the migrated cell counts using the Image J Software.

In addition, we have carefully checked our manuscript and made corrections in revised tracked manuscript.

We hope meet with approval!

Best regards!

This manuscript is a resubmission of an earlier submission. The following is a list of the peer review reports and author responses from that submission.

Round 1

Reviewer 1 Report

The study by Xue etal shows that secreted hsp90 promotes EMT in pancreatic cancer and that targeting secreted HSP90-LRP1 axis is beneficial in enhancing chemotherapy response of the cells. The study is interesting but authors should address the following major concerns:-

  1. Secreted HSP90 results are not convincing. Authors should conduct ELISA to actually measure secreted HSP90. Authors mention secreted HSP90 was more as compared to surface HSp90. However, western blot analysis shows surface HSP90 is equally expressed as secreted form. So how did authors come to this conclusion. Also, why did authors chose b-actin as loading control for secreted HSP90? They should include ponceau/commassie stain for equal loading. Data presented is only with one cell line Capan2. To increase robustness authors should include data from atleast one more invasive cell line.
  2. What is ‘control’ in wound healing assay? Does anti-HSP90 antibody bind to surface HSP90 also? Please quantify the results from this assay. How many replicates were done? Similarly, how many replicates of invasion assay were performed? How did authors measure invasion. Please present the graph comparing different groups.
  3. In figure 3, authors should include results from CM+ anti HSP90 group to truly point out the role of secreted hsp90 in EMT.
  4. Figure 5, results show LRP inhibition decreases EMT. However the role of secreted HSP90 in facilitating LRP driven downstream signaling is weakly presented. mRNA expression of EMT markers in rhsp90 alone should also be presented on the graph. Protein analysis on AKT/ERK signaling should be done in presence of shLRP1 with or without rhsp90/CM.
  5. Figure 6, authors should conduct drug sensitivity assays in one more cell line. Why did authors chose doxorubicin and topotecan?

 Minor comments

  1. Introduction: 5 year survival rate reference is not updated. The new overall 5 year survival rate is 11%. This should be updated with new reference.
  2. The manuscript overall needs major proofreading and editing
  3. There are major spelling errors throughout the manuscript
  4. The methods section needs more details. For example wound healing assay should be quantitated and details how the wound was measured is missing.
  5. Western blot analysis methods mention band intensities were quantitated. However relative numbers of band intensities are missing in the figures. How was secreted HSP90 normalized? B-actin is not the right control. Authors should include commassie stain/ponceau stain of equal loading.

Author Response

Dear Reviewer:

We are honored to get your support and appreciation of our work! Thank you very much for these precious comments concerning my manuscript entitled “Secreted HSP90α-LRP1 signaling promotes tumor metastasis and chemoresistance in pancreatic cancer” (ijms-1622054).

These comments are all valuable and very helpful for revising and improving my paper, as well as the important guiding significance to my researches. We have studied comments carefully and have addressed each point by reviewer below.

We hope meet with approval!

Reviewer 2 Report

The manuscript by Xue and colleagues, present data on the role of secreted HSP90α on the invasiveness of pancreatic cells and in their metastatic transition.

Overall the research conducted is sound and the discussion is more or less coherent with the presented results (both in the main text and in the supplementary material). However the manuscript could benefit from a more rational way of presenting the results. The Authors mix the results on individual PL45 and Capan2 tumor lineange. For a better understanding, a separation between all the experiments on PL45 and all the experiments on Capan2 would be more beneficial. For example, Figure 3B-C-D should be only on Capan2, while another 3 panels only on PL45.

In figure 1A there are 2 isoforms of eHSP90a, the one with the lower MW has a very similar pattern of the secreted one in Fig 1B -> the Authors should perform a mass spectrometry (MS) analysis of the two bands in Fig 1A and of that in Fig 1B before concluding on the role of the membrane protein vs secreted protein.

In Figure 3C and 4A a different zoom/magnification should be used to appreciate the morphological change stated in the text. In the western blot experiments of Figure 3: why the antibody against e-caderin has been used if the results from transcription show that n-caderin is 25-fold more increased?

Figure 5A and 5B and text line 167: there is a correlation between mRNA of LRP1 and snail, but it is not significant (0.6 is very close to the random threashold).

Figure 6 and paragraph 2.6: only gemcitabine has a positive synergistic effect in combination with LRP1 silencing. doxorubicin has an opposing effect (increasing mortality over control), while the other 3 chemotherapeutics have no effect. By the way, the list of abbreviation is not exaustive and not all the acronyms used throughout the text are listed.

The discussion paragraph needs to be rewritten since many sentences lack the verb or have grammar errors, leading to obscure text, which also refers to data not shown. Moreover, the sentence in lines 219-221 should have been put before paragraph 2.3 in order for the reader to understand why the Authors had chosen only 2 markers (vimentin and e-caderin).

Minor points:

line 38: "in PC is mainly..."

line 52: "in physiological conditions"

line 59: "interacting"

line 63: "Thuringer"

lines 68-69: what is the meaning of "elite"? I do not understand the sentence

line 165: delete AND before Correlation

line 248: "ERK activation"

Author Response

(The authors gave the same response as above.)

Round 2

Reviewer 1 Report

Major concerns still remain and manuscript shouldn't be accepted in its present form

  1. Inclusion of data from one more invasive cell line. This was asked after first review but the authors have only added one uncovincing data without stats (wound healing) that doesn't add to the rigor.
  2. Authors have not included a treatment group as asked in Fig 3. They have added morphology pictures from CM+ Anti-HSP90 group which are not convincing at all. They should include data from this group in panels C-E.
  3. Flowcytometry data needs more explanation and the method should be described in the methods section.
  4. LRP-HSP90 connection is very weak as according to Fig 5e rhsp90 in presence of shLRP increases EMT markers which means HSP90 doesnot require LRP. A rhsp90 alone group is still missing.
  5. Fig 5J. Protein levels of pAKT/AKT seem to be downregulated in western blot and according to the text this should be low with shLRP. However, the new graph shows increase in pAKT. This questions how these data were analyzed?
  6. Figure panels are not presented in a systematic order. That needs to be corrected.

Reviewer 2 Report

The Authors have followed whenever possible the suggestions and have explained the points which were unclear in the first version. The manuscript has indeed improved.